# Characterization and Phylogenetic Analysis of the Complete Mitochondrial Genome of *Triplophysa microphthalma*

**DOI:** 10.3390/biology13080608

**Published:** 2024-08-11

**Authors:** Ping Yang, Wei Guo, Chao Wei, Xin Wang, Yixuan Wang, Jia Wang

**Affiliations:** Xinjiang Key Laboratory for Ecological Adaptation and Evolution of Extreme Environment Biology, College of Life Sciences, Xinjiang Agricultural University, Urumqi 830052, China; zkayp1314@163.com (P.Y.); guowei612@xjau.edu.cn (W.G.); 13095205790@163.com (C.W.); 18146913651@163.com (X.W.); 17709908643@163.com (Y.W.)

**Keywords:** *Triplophysa microphthalma*, characterization, phylogenetic, complete mitochondrial genome

## Abstract

**Simple Summary:**

The fine-eyed plateau loach, *Triplophysa microphthalma*, is a species of the genus *Triplophysa* in the phylum Chordata. According to the Catalogue of Life China (2024), it is primarily found in the Xinjiang Urumqi Autonomous Region of China. This species has an elongated, cylindrical body that is grayish-brown or yellowish-brown and marked with spots or stripes, aiding in camouflage in the underwater environment. Its smaller eyes, shorter dorsal and caudal fins, and longer ventral and pectoral fins are adaptations that make it well-suited for bottom activities. The lower-positioned mouth is adapted for a benthic lifestyle, facilitating food foraging at the bottom of the body of water. The presence of barbels aids in foraging in turbid waters. The taxonomic classification of *T. microphthalma* has been contentious, with its placement within either the *Triplophysa* or *Barbatula* genus influencing our understanding of its biological characteristics. In this study, we report the complete mitogenome of *T. microphthalma* and conduct a detailed analysis of its characteristics to infer phylogenetic relationships. Collectively, our research clarified the genus-level phylogenetic relationships of this species.

**Abstract:**

The complete mitochondrial genome has been extensively utilized in studies related to phylogenetics, offering valuable perspectives on evolutionary relationships. The mitochondrial genome of the fine-eyed plateau loach, *Triplophysa microphthalma*, has not attracted much attention, although this species is endemic to China. In this study, we characterized the mitochondrial genome of *T. microphthalma* and reassessed the classification status of its genus. The complete mitochondrial genome of *T. microphthalma* was 16,591 bp and contained thirty-seven genes, including thirteen protein-coding genes (PCGs), two ribosomal RNA genes (rRNAs), and twenty-two transfer RNA genes (tRNAs). All but one of the thirteen PCGs had the regular start codon ATG; the gene *cox1* started with GTG. Six PCGs had incomplete stop codons (T--). These thirteen PCGs are thought to have evolved under purifying selection, and the mitogenome shared a high degree of similarity with the genomes of species within the genus *Leptobotia*. All tRNA genes exhibited the standard clover-shaped structure, with the exception of the trnS1 gene, which lacked a DHU stem. A phylogenetic analysis indicated that *T. microphthalma* was more closely related to species within the genus *Triplophysa* than to those in *Barbatula*. The present study contributes valuable genomic information for *T. microphthalma*, and offers new perspectives on the phylogenetic relationships among species of *Triplophysa* and *Barbatula*. The findings also provide essential data that can inform the management and conservation strategies for *T. microphthalma* and other species of *Triplophysa* and *Barbatula*.

## 1. Introduction

The family Coboitidae, under the order Cypriniformes, was originally proposed by Regan [1]. It comprises the Nemacheilinae, Botiinae, and Coboitinae subfamilies. Among these, Nemacheilinae boasts a diverse array of species that are widely distributed across the Eurasian continent, with some even reaching parts of Northern Africa. The fine-eyed plateau loach, commonly known as the small-eyed whiskered loach, is found in cold water streams and ditches with gravelly bottoms in Hami City, Turpan City, and Toksun County in Xinjiang, China. China is home to the highest number of loach genera and species, many of which are unique. The fine-eyed plateau loach has a complex taxonomic history, with its scientific name changing several times. It was initially classified as *Diplophysa microphthalma*, then as *Barbatula microphthalma* [2], followed by *Labiatophysa microphthalma*, finally being confirmed as *Triplophysa microphthalma* [3]. These species of the Coboitinae subfamily are mainly found in natural water habitats in regions such as Hami, Turpan, Urumqi, and Bortala Mongol Autonomous Prefecture in Xinjiang, China [4]. Currently, there is limited research on this species, with studies primarily focusing on morphological descriptions. The debate on whether this species belongs to the genus *Triplophysa* or *Barbatula* remains unresolved, with research still existing at a superficial level [5].

Mitochondria, which play essential roles in energy metabolism, genetic signaling, biological synthesis, and cell apoptosis [6], are widely found in eukaryotic cells. It has been established that animal mitochondria possess an independent genome with distinct characteristics including maternal inheritance, a simple structure, a high rate of evolution, and ease of amplification [7,8,9]. With the rapid advancement of sequencing technology and molecular biotechnology, the mitochondrial genome has garnered significant interest. Furthermore, mitochondria have been extensively employed in research pertaining to species evolution, phylogenetic relationships, and population genetics, emerging as a potent tool for elucidating genetic variance among closely related species and populations, particularly among vertebrates like amphibians [10], reptiles [11], and birds [12,13]. Additionally, there has been a significant increase in the sequencing of fish mitochondrial genomes. For instance, investigations into gene arrangements, tRNA secondary configurations, and transcription and replication regions within fish mitochondrial genomes have yielded a plethora of dependable data on fish taxonomy, ecological adjustments, and evolutionary lineages. This underscores the significance of complete mitochondrial genomes as crucial molecular assets for the preservation and sustainable management of fish populations [14].

In this study, we sequenced, assembled, and characterized the complete mitochondrial genome of *T. microphthalma* for the first time, and also compared its structures and the complete mitochondrial genome with those of some Cypriniformes species. Our aim was to comprehensively understand the complete mitochondrial genome of *T. microphthalma* and assess its phylogenetic position. Therefore, the results of this study provide fundamental molecular data for the conservation and management of *T. microphthalma* and contribute to a deeper understanding of the phylogenetic relationships within Cobitidae.

## 2. Materials and Methods

### 2.1. Sample Collection and DNA Extraction

The body of *T. microphthalma* is naked and scale-free. The anterior and posterior nostrils are clearly separated. Specimens for this study were collected from the Poplar River at coordinates 42.995706 N, 88.741019 E, located in Urumqi, Xinjiang, China. The specimens were brought back to the laboratory of the College of Life Sciences, Xinjiang Agricultural University. The *T. microphthalma* specimens were executed in anhydrous ethanol and subsequently dissected, and the intact liver was removed and preserved in anhydrous ethanol for subsequent genomic DNA extraction. In this study, experiments on *T. microphthalma* were conducted in compliance with the National Guidelines for the Care and Treatment of Laboratory Animals. Liver DNA of the *T. microphthalma* specimens was extract using an animal tissue DNA extraction kit from Chengdu Fuji Biotechnology Co., Ltd. (Chengdu, China) and the quality of the genomic DNA was assessed using 0.8% agarose gel electrophoresis. The DNA samples were sent to Sangon Biotech Co., Ltd. (Shanghai, China) for storage at minus 20 degrees Celsius.

### 2.2. Genome Sequencing and Assembly

In this study, we obtained the complete mitochondrial genome sequence of *T. microphthalma* using second-generation sequencing (NGS) technology. Library construction was performed using the Illumina 12200ES DNA Library Prep Kit reagent. Then, mitochondrial genome sequencing was performed based on the DNBSEQ T7(v3) platform of Shenzhen Huada Genetics Co. Moreover, FASTQ was used to filter the raw sequencing data, remove the splices and low-quality sequences, and obtain clean reads for quality control [15].

Mitochondrial genome assembly was initially performed using SPAdes for clean reads [16]. Then, BLAST was used to compare the scaffolds with the Nucleotide Sequence Database for sequence similarity information, extracting sequencing depth and coverage information for each scaffold. Finally, possible targets were manually selected after organizing and comprehensively considering the above information scaffolds. The contigs obtained using splicing were then used to fill the gaps in the sequence scaffolds using GapFiller [17]. Finally, PrInSeS-G was used for sequence correction to address editing errors during splicing and insertion deletions of small fragments until the complete pseudo-genome sequence was assembled [18]. The sequences were further processed using built-in scripts to evaluate and process the assembly results for cyclization to obtain the final complete cyclized genome.

### 2.3. Sequence Annotation and Analysis

We used BLASTN with GeneWise to obtain CDS gene boundaries by reverse matching with near-origin reference databases including MiTFi to obtain tRNA sequence annotations and Rfam’s cmsearch to identify non-coding rRNAs by comparison [19], which were finally summarized and organized into complete annotation results. The tRNAs of the *T. microphthalma* were predicted on the server Ubuntu (version 22.04.4) using the software tRNAscan-SE (version 2.0) and then plotted at the online website (http://cloud.genepioneer.com:9929/#/tool/alltool?type=4, accessed on 8 May 2024) [20]. Circular mitochondrial genome mapping was generated by the Circos (version 0.69-9) software [21]. Relative synonymous codon usage, initiation and termination codons, and the nucleotide composition of protein-coding genes were analyzed with the software PhyloSuite (version 1.2.3), and codon preference maps (RSCUs) were drawn with RStudio (version 4.4.2) [22]. The complete mitochondrial genome sequence of *T. microphthalma* has been submitted to the GenBank with the accession number PP979136.

### 2.4. Phylogenetic Analysis

To clarify the phylogenetic position of the fine-eyed plateau loach and to understand the phylogenetic relationships of the Cyprinidae family, the mitochondrial genome sequences of 43 Cyprinidae species were downloaded from the GenBank database (https://www.ncbi.nlm.nih.gov/, accessed on 25 April 2024) (Table 1). Phylogenetic trees were reconstructed using the nucleotide sequences of 13 PCGs, 22 tRNAs, and 2 rRNAs from the mitochondrial genomes of all species (43 species including *T. microphthalma*).

The phylogenetic tree was reconstructed using the complete mitochondrial genome sequences of 43 Cypriniformes species, with *Leptobotia Pellegrini*, *Leptobotia Pellegrini*, *Leptobotia taeniops*, and *Leptobotia rubrilabris* as outgroups (Table 1). All genome sequences were aligned to the same starting point in the circular mitochondrial genome. In this study, the mitochondrial genomes of 43 species were firstly subjected to multiple sequence comparisons using MUSCLE (version 3.8.31), and the appropriate nucleotide substitution models were selected by trimming the compared sequences using the software trimAL (version 1.5.0) [23]. Subsequently, the RaxML (version 8.2.12) software was used for a rapid bootstrap analysis with 1000 bootstrap replicates to construct the maximum likelihood phylogenetic tree [24]. The alternative model used the General Time Reversible (GTR) model, and the rate heterogeneity model used the GAMMA model and the BFGS method (Broyden–Fletcher–Goldfarb–Shanno) to optimize the GTR rate parameters, and the tree file was imported into the Interactive Tree Of Life (iTOL) website (embl.de) for landscaping. The near-origin genomes were compared using MUMmer (version 4.0.0) to obtain the best matching regions and mapping [25]. A covariance analysis of *T. microphthalma* with closely related species using the same parameters was used to determine the taxonomic status of *T. microphthalma***.**

## 3. Results 

### 3.1. Mitochondrial Structural Characteristics

The mitochondrial genome of *T. microphthalma*, obtained using high-throughput sequencing technology, forms a closed circle with a total length of 16,596 base pairs (Figure 1). It comprises 13 typical protein-coding genes (PGCs), 2 ribosomal RNA genes (rRNAs), and 22 transporter RNA genes (tRNAs). Among the mitochondrial genes, nine genes, trnQ, trnA, trnN, trnC, trnY, trnS2, nad6, trnE, and trnP, are encoded by the light (L) strand, and the rest of the genes are encoded by the heavy (H) strand. This is similar to previous findings on the mitochondrial genome of fish [26,27,28]. All PCGs start with the initiation codon ATG, except *cox1*, which starts with GTG. Seven genes use the complete termination codon TAA, while six have incomplete termination codons (TA-, GA-, or T--) (Table 2). The mitochondrial genome features an intergenic overlap of 26 bp, with the longest stretch located between the *atp8* and *atp6* genes (Table 2). Additionally, 13 intergenic spacer regions of 68 bp in total length were identified in the mitochondrial genome, varying in size from 1 to 31 bp, with the largest spacer found between the trnN and trnC genes (Table 2).

### 3.2. Protein-Coding Genes and Codon Usage

PCGs accounted for 68.91% of the total length of the mitochondrial genome of *T. microphthalma* (Table 3). The whole base composition of the mitochondrial genes of *T. microphthalma* (Table 3) was as follows: 28.2% T, 25.7% C, 28.1% A, 18.0% G, 56.3% AT, and 43.7% GC. The analysis of the nucleotide composition indicated that the A + T content of the mitochondrial genome (56.3%) was greater than the G + C content (43.7%). Additionally, within different gene types, such as tRNAs and rRNAs, the A + T content was also higher than the G + C content, with tRNAs showing an A + T content of 56.3% and rRNAs showing 54.4%. These results suggest a nucleotide composition bias towards A + T in *T. microphthalma*’s mitochondrial genome (Table 3). The AT-skew and GC-skew values across the mitochondrial genome were −0.003 and −0.177, respectively, implying a higher abundance of A over T and C over G (Table 3). Furthermore, codon usage in the PCGs favored A- and C-rich codons.

### 3.3. Transporter RNA, Ribosomal RNA, and Protein-Coding Gene Codon Usage

The complete mitochondrial genome of *T. microphthalma* included 22 tRNAs and 2 rRNAs. The tRNAs varied in size from 66 to 75 bp, totaling 1558 bp (Table 2). The tRNAs were positive on the AT (0.009) and GC (0.050) biases, suggesting that the tRNAs had a slight bias towards A and G (Table 3). The resulting inferred secondary structure of tRNAs supports the idea that 21 out of 22 tRNAs can fold into a typical cloverleaf structure consisting of a D-loop, a TΨC loop, a D-arm, an anticodon loop, and a T-arm (Figure 2, in which trnS1, i.e., Ser1, displays the distinguishing feature of a deletion of the DHU arm, which is replaced by a loop, as is the case in the other spirochetes [29].

The 13 PCGs of the mitochondrial genome encode a total of 3806 amino acids. All PCGs initiate with either ATG or GTG, with 12 of the 13 PCGs using ATG and only *cox1* using GTG (Table 2). The termination codons are expressed as TAA, TAG, and GA single T. There was one PCG (*nad5*) with TAG as the termination codon, one PCG (*nad3*) with GA as the termination codon, one PGC (*nad4*) with TA as the termination codon, and four PCGs (*nad2*, *cox2*, *cox3*, and *cytb*) with a single T as the termination codon. The remaining six PCGs (*nad1*, *cox1*, *atp8*, *atp6*, *nad4L*, and *nad6*) were terminated with a TAA termination codon (Table 2). The relative synonymous codon usage (RSCU) values for these 13 PGCs are depicted in Figure 3. For the fine-eyed plateau loach, the most frequently used codons are CGA (Arg), TTA (Leu), TCA (Ser), and GCC (Ala). Codon usage of the PCGs is biased towards A- and C-rich codons, and this usage varies between species [30].

The 16S and 23S rRNAs, encoded by the rrnS and rrnL genes, respectively, are both located on the heavy strand (H) and are 952 bp and 1675 bp in length (Table 2). These rRNA genes are situated between the trnF and trnL2 genes, with the trnV gene interspersed between the rRNAs. The rRNAs exhibit a higher A + T content, which is 54.4%. They have a positive AT-skew value of 0.203 and a slightly negative GC-skew value of -0.031 (Table 3). Additionally, the mitochondrial genome of *T. microphthalma* also includes a control region, which is 916 bp in length.

### 3.4. Phylogenetic Relationships

In this study, we used the maximum likelihood method to draw a phylogenetic tree based on the tandem nucleotide sequences of 13 PCGs, 22 tRNAs, and 2 rRNAs from the mitochondrial genome, and we determined the phylogenetic relationships belonging to a total of 43 species in 11 genera. The phylogenetic tree showed a high level of support for most nodes, with a bootstrap proportion of ≥95.65%. However, some branches had nodes with a relatively low level of support; for example, the node with the lowest support had a bootstrap value of 4.34%, which is less than 50%. The 11 genera selected in the present study were *Sinibotia*, *Leptobotia*, *Parabotia*, *Chromobotia*, *Botia*, *Yasuhikotakia*, *Pangio*, *Cobitis*, *Acanthocobitis*, *Barbatula*, and *Triplophysa*. *B. microphthalma* was categorized as a genus of loach in previous studies; however, based on the results of the present study, it is more closely related to the species of the genus *Platyrhynchos* and more distantly related to another species of loach, *B. barbatula* (Figure 4).

Our findings reinforce the fact that *T. microphthalma* should be categorized as a genus *Triplophysa*, since *T. microphthalma* is more closely related to species of the genus *Triplophysa*. Since the categorization of *T. microphthalma* has not been uniform in previous studies [3] in order to further determine the proximity of *T. microphthalma* to the species of the loach genus *Triplophysa* and further verify the reliability of our results, we chose *B. barbatula* and the three species that clustered closest to *T. microphthalma* on the phylogenetic tree to be covariates with *T. microphthalma* and conducted a covariate analysis using the same parameters (Figure 5). The results of the covariance analysis showed that *T. microphthalma* had fewer conserved regions of mitochondrial genome covariance with *B. barbatula* (Figure 5a). Moreover, in contrast to *B. microphthalma* with *Triplophysa ulacholica* (Figure 5b), *Triplophysa strauchii* (Figure 5c) and *Triplophysa dorsalis* (Figure 5d) had more co-linear conserved regions of mitochondrial genes. This suggests that the mitochondrial genome of *T. microphthalma* is closer to that of species in the genus *Triplophysa*, which further supports the reliability of our experimental results.

## 4. Discussion

This study presents the first characterization of the complete mitochondrial genome of *T. microphthalma*. Our results indicate that its mitochondrial genome is a closed circular structure with a total size of 16,569 bp and is composed of thirty-seven genes (thirteen PCGs, twenty-two tRNAs, and two rRNAs). The genome organization and composition of *T. microphthalma* are similar to those of most fish species reported previously [31,32]. The content of AT was 56.3%, indicating the existence of AT preference. This is similar to the base preference of almost all postnatal animals, including fishes, which is consistent with the evolutionary characteristics of mitochondrial DNA. This suggests that the AT-enriched region may correspond to the replication origin and be more prone to change during evolution [33,34]. Of course, there may be slight variations in the skewness of AT and GC among different species, which can serve as a reference in the evaluation of the phylogenetic status and relationships with identified species.

The results of our investigation showed that the PCGs encompass a significant portion of the mitochondrial genome of *T. microphthalma*, amounting to 68.91% of the total length. The results indicate that most PCGs initiate with ATG, except for *cox1*, which initiates with GTG, a pattern consistent with observations in other fish species, such as *Schizothorax macropogon* and *Leptobotia elongata* [1,35]. Seven genes use the complete stop codon TAA or TAG, while six genes use incomplete stop codons (TA-, GA-, or T--), a pattern common among teleosteans, as reported in a number of studies including those on *L. elongata* [1], *Cobitis macrostigma* [36], and *Parabotia kiangsiensis* [37]. The tRNA sequences located at the 3′ end of these genes may lead to incomplete stop codons, which could potentially be modified to TAA through post-transcriptional polyadenylation [30]. Among the 22 tRNA genes, 21 genes are known to fold into a typical clover structure, while tRNASer (AGY) is observed to lack a D-stem. It has been suggested that this abnormal tRNA structure may play a similar role to normal tRNA by adjusting its structural conformation [38]. At the same time, some tRNA genes have been reported to exhibit non-classical pairing phenomena such as A-A, C-U, A-G, U-U, and A-C, and some tRNA mismatches are commonly observed in other animals, which may contribute to the maintenance of tRNA secondary structure stability [39,40]. However, only a C-U mismatch was observed in the tRNAs of *T. microphthalma* in this study.

The taxonomic placement of *T. microphthalma* within the genus has been a subject of debate. While some researchers classify it under the genus *Triplophysa*, it is still categorized under the *Barbatula* genus in the China Animal Scientific Database. A phylogenetic tree was developed utilizing the complete mitochondrial genome sequences from *T. microphthalma* and 43 published species of Cobitoidea. The results indicated strong support for placing *T. microphthalma* within the *Triplophysa* genus, rather than *Barbatula*, with a high level of confidence. Additionally, the phylogenetic tree showed that Cobitoidea can be divided into five groups. In other studies, Cobitoidea has also been divided into five groups: Balitoridae, Nemacheilidae, Cobitidae, Botiidae, and Vaillantellidae [41]. In previous studies, for the phylogenetic tree analysis, there were protein-coding genes analyzed [36], and, usefully, the entire mitochondrial genome was analyzed [42]. This may result in a difference in their taxonomic status. The discrepancy in our classification outcome could stem from the complete mitochondrial genome analysis chosen for analysis. Furthermore, the phylogenetic tree revealed that *Barbatula* and *Triplophysa* form a sister group, which in turn is a sister group to *Acanthocabitis*. These genera belong to Nemacheilidae in the classification based on morphology, indicating that the results of mitochondrial evolution are consistent with morphological classification. Other studies have indicated that both adaptive evolution and interbreeding among closely related species can also influence inferred phylogenetic relationships [43,44].

The species Leptobotia mantschurica and Sinibotia reevesae belong to the genera Leptobotia and Sinibotia, respectively, but these two species clustered into a single unit in our evolutionary tree, yet they did not cluster with the species of this genus, which may be taxonomically controversial as well. However, they were not studied further since they were not the subject of this paper. We will continue to monitor the taxonomy of these two species in subsequent studies.

Relying solely on morphological or phylogenetic methods for classification can lead to biases, as drawing phylogenetic relationships from limited markers is not always precise. Hence, a comprehensive approach involving an increased number of samples and sequence data is essential to establish a more robust phylogenetic framework. This approach should also integrate morphological characteristics to accurately ascertain the status and relationships within the taxonomic classification. To further verify the reliability of our results, we selected *T. ulacholica* and *T. dorsalis* to conduct a covariance analysis with *T. microphthalma*. The analysis results showed that the covariance level of the mitochondrial genomes of these three species was high, indicating that the differentiation time between these three species was short and the accumulated variation was low. Therefore, the homology of the mitochondrial genomes of the three species was high, and the differentiation of the three species had a high degree of conservatism. Our results indicate that the fine-eyed plateau loach is more homologous to the plateau loach and, therefore, should be categorized in the genus *Triplophysa*.

## 5. Conclusions

This study reported the first complete mitogenome of *T. microphthalma*. The mitochondrial genome, with a total size of 16,569 bp, comprised thirty-seven genes (thirteen PCGs, twenty-two tRNAs, and two rRNAs). All 13 PCGs, except for the *cox1* gene, which started with GTG, exhibited the regular start codon ATG. All tRNA genes had standard cloverleaf structures, except for *trnS1*, which lacked a DHU stem. A phylogenetic analysis indicated that *T. microphthalma* was more closely related to the genus *Triplophysa* than to *Barbatula*. Overall, this study contributes to a deeper understanding of the mitochondrial genome structure and phylogenetics of *T. microphthalma*. However, detailed information about many species within the Cobitoidae family remains elusive. Therefore, it is recommended that future phylogenetic studies of Cobitoidae incorporate a broader range of taxa to provide a more comprehensive understanding of their evolutionary history.

## Figures and Tables

**Figure 1 biology-13-00608-f001:**
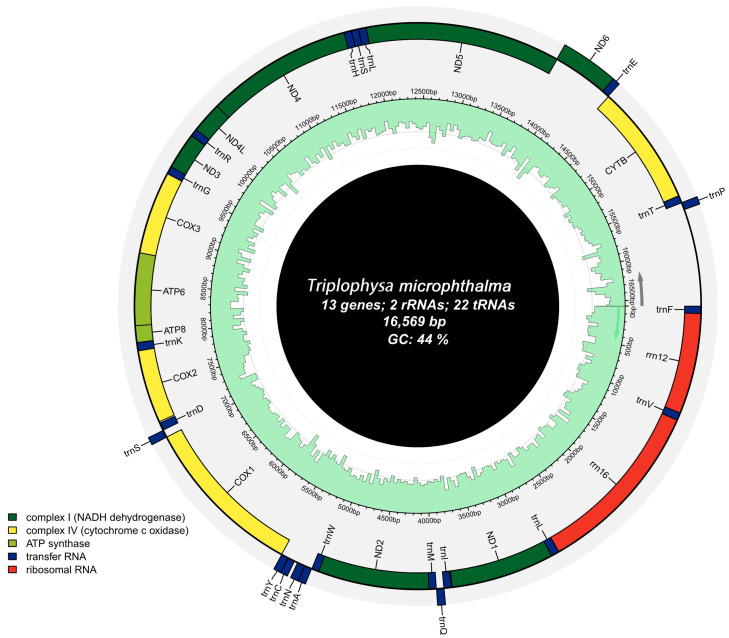
Mapping of the mitochondrial genome of *Triplophysa microphthalma*.

**Figure 2 biology-13-00608-f002:**
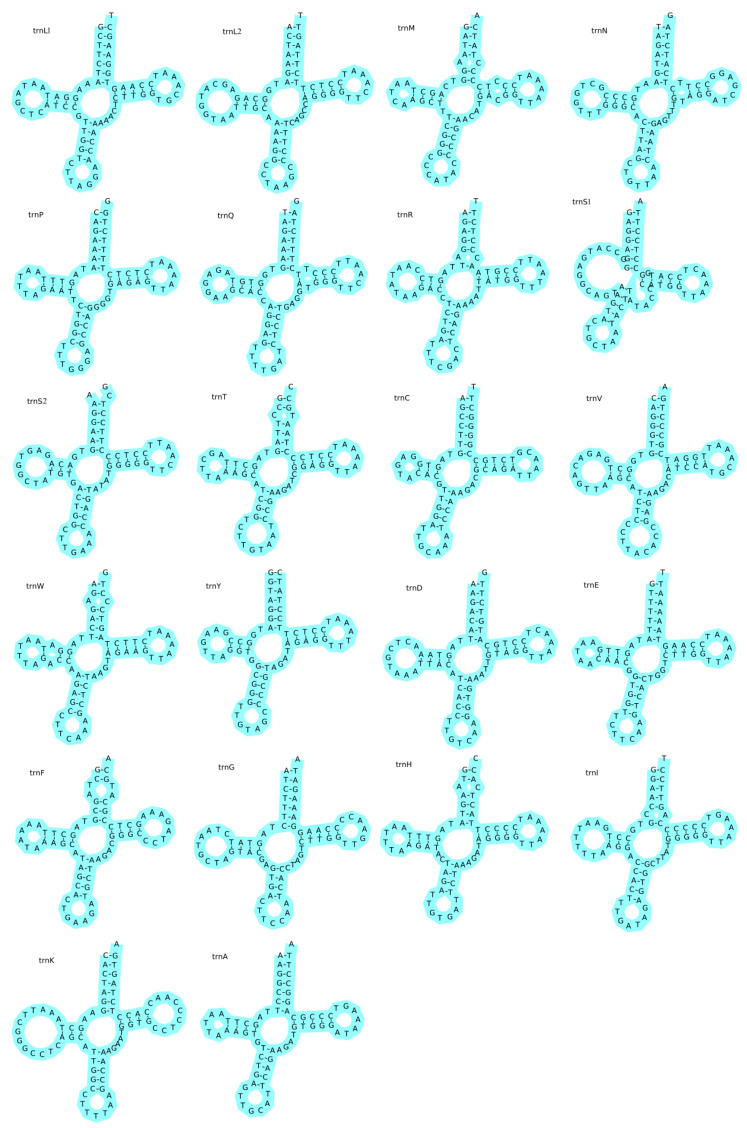
The predicted secondary structures of the 22 tRNAs within the mitochondrial genome of *Triplophysa microphthalma*. Each tRNA gene is identified by its standard abbreviations in the upper left corner, alongside its corresponding amino acid in parentheses for translation purposes.

**Figure 3 biology-13-00608-f003:**
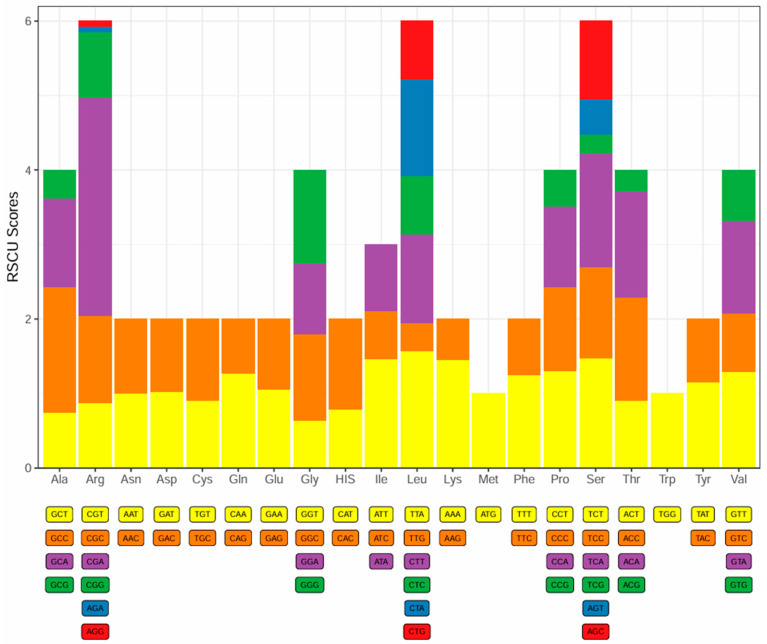
The relative synonymous codon usage (RSCU) values for PCGs in the complete mitochondrial genome of *Triplophysa microphthalma*. Distinct colors signify various codon families that correspond to amino acids.

**Figure 4 biology-13-00608-f004:**
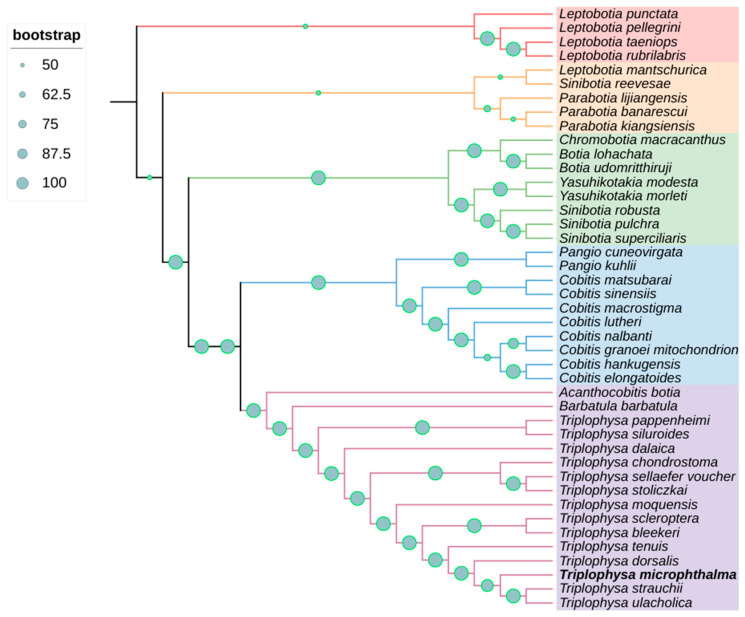
A maximum likelihood (ML) phylogenetic tree, bolstered by bootstrap support values at the nodes, was constructed using nucleotide sequences from 43 Cyprinidae species, encompassing 13 PCGs, 22 tRNAs, and 2 rRNAs.

**Figure 5 biology-13-00608-f005:**
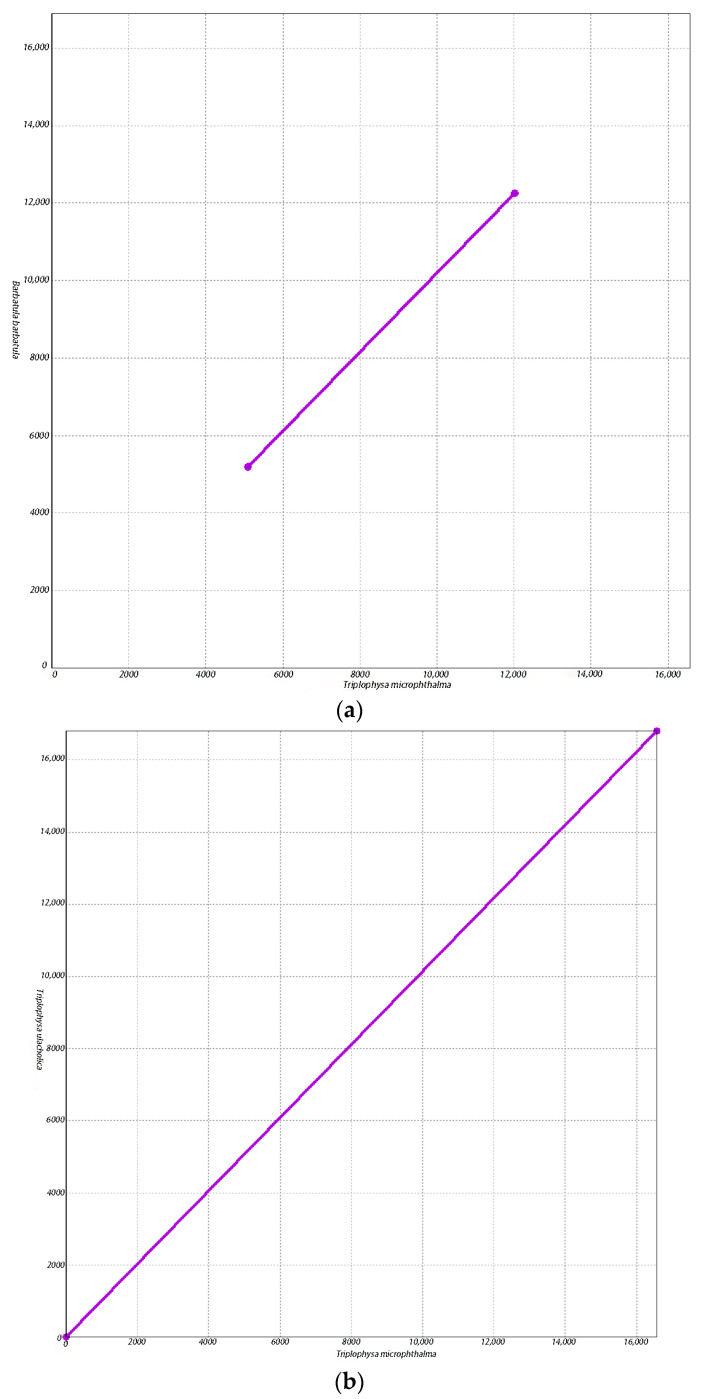
Co-linear scatter plot. The horizontal and vertical axes represent two different genomes, and the connected portion of the line represents the region of covariance conserved between the two genomes. The more the lines are connected, the more similarities exist between the two genomes. (**a**) Co-linear scatter plot of *Triplophysa microphthalma* and Barbatula barbatula. (**b**) Co-linear scatter plot of *Triplophysa microphthalma* and *Triplophysa ulacholica*. (**c**) Co-linear scatter plot of *Triplophysa microphthalma* and Triplophysa strauchii. (**d**) Co-linear scatter plot of *Triplophysa microphthalma* and Triplophysa dorsalis.

**Table 1 biology-13-00608-t001:** The species rosters utilized in the phylogenetic analyses of this study.

Family	Genus	Species	Assession Number
Cobitidae	Sinibotia	*Sinibotia reevesae*	NC 030322.1
		*Sinibotia robusta*	NC 027853.1
		*Sinibotia pulchra*	KT362179.1
		*Sinibotia superciliaris*	JX683724.1
	Leptobotia	*Leptobotia mantschurica*	AB242170.1
		*Leptobotia elongata*	OR818399.1
		*Leptobotia microphthalma*	KY307846.1
		*Leptobotia punctata*	MH644033.1
		*Leptobotia pellegrini*	NC 031602.1
		*Leptobotia taeniops*	AP013304.1
		*Leptobotia rubrilabris*	KY307847.1
	Parabotia	*Parabotia lijiangensis*	MT323118.1
		*Parabotia banarescui*	NC 026127.1
		*Parabotia kiangsiensis*	MT850132.1
	Chromobotia	*Chromobotia macracanthus*	AB242163.1
	Botia	*Botia lohachata*	KP729183.1
		*Botia udomritthiruji*	AP011349.1
	Yasuhikotakia	*Yasuhikotakia modesta*	KY131962.1
		*Yasuhikotakia morleti*	NC 031600.1
	Pangio	*Pangio cuneovirgata*	NC 031594.1
		*Pangio kuhlii*	NC 031599.1
	Cobitis	*Cobitis matsubarai*	NC 029441.1
		*Cobitis sinensiis*	NC 007229.1
		*Cobitis macrostigma*	MK156771.1
		*Cobitis lutheri*	NC 022717.1
		*Cobitis nalbanti*	MH349461.1
		*Cobitis granoei mitochondrion*	NC 023473.1
		*Cobitis hankugensis*	MN841275.1
		*Cobitis elongatoides*	NC 023947.1
Nemacheilidae	Acanthocobitis	*Acanthocobitis botia*	AP012139.1
	Barbatula	*Barbatula barbatula*	CM074126.1
	Triplophysa	*Triplophysa microphthalma*	PP979136
		*Triplophysa pappenheimi*	NC 033972.1
		*Triplophysa siluroides*	NC 024611.1
		*Triplophysa dalaica*	KT213590.1
		*Triplophysa chondrostoma*	KT213589.1
		*Triplophysa sellaefer voucher*	KY851112.1
		*Triplophysa stoliczkai*	JQ663847.1
		*Triplophysa moquensis*	KT213597.1
		*Triplophysa scleroptera*	KT213602.1
		*Triplophysa bleekeri*	JQ686729.1
		*Triplophysa tenuis*	KT224363.1
		*Triplophysa dorsalis*	KT213591.1
		*Triplophysa strauchii*	KP979754.1
		*Triplophysa ulacholica*	KT259194.1

**Table 2 biology-13-00608-t002:** Summary of the *Triplophysa microphthalma* loach mitogenome.

Gene	Position		Size	Intergenic Nucleotides	Codon		Strand
	From	To			Start	Stop	
trnF	1	69	69	0			H
rrnS	70	1021	952	0			H
trnV	1022	1093	72	0			H
rrnL	1094	2768	1675	0			H
trnL2	2771	2845	75	2			H
nad1	2846	3820	975	0	ATG	TAA	H
trnI	3828	3899	72	7			H
trnQ	3898	3968	71	−2			L
trnM	3970	4038	69	1			H
nad2	4039	5083	1045	0	ATG	T--	H
trnW	5084	5153	70	0			H
trnA	5156	5224	69	2			L
trnN	5226	5298	73	1			L
trnC	5330	5395	66	31			L
trnY	5396	5463	68	0			L
cox1	5465	7015	1551	1	GTG	TAA	H
trnS2	7016	7086	71	0			L
trnD	7089	7161	73	2			H
cox2	7175	7865	691	13	ATG	T--	H
trnK	7866	7941	76	0			H
atp8	7943	8110	168	1	ATG	TAA	H
atp6	8101	8784	684	−10	ATG	TAA	H
cox3	8784	9567	784	−1	ATG	T--	H
trnG	9568	9640	73	0			H
nad3	9641	9987	347	0	ATG	GA-	H
trnR	9990	10,059	70	2			H
nad4L	10,060	10,356	297	0	ATG	TAA	H
nad4	10,350	11,731	1382	−7	ATG	TA-	H
trnH	11,732	11,801	70	0			H
trnS1	11,802	11,869	68	0			H
trnL1	11,871	11,943	73	1			H
nad5	11,944	13,782	1839	0	ATG	TAG	H
nad6	13,779	14,300	522	−4	ATG	TAA	L
trnE	14,301	14,369	69	0			L
cytb	14,374	15,514	1141	4	ATG	T--	H
trnT	15,515	15,585	71	0			H
trnP	15,584	15,653	70	−2			L
D-loop	15,654	16,569	916				H
Overlap:	6	Gap:	13				

Note: “H” represents the heavy strand; “L” represents the light strand.

**Table 3 biology-13-00608-t003:** Nucleotide composition and skewness for the heavy (H) and light (L) strands of the *Triplophysa microphthalma* loach mitochondrial genome.

Regions	Size (bp)	T (U)	C	A	G	AT (%)	GC (%)	GT (%)	AT Skew	GC Skew
PCGs	11,418	30.3	26.2	25.5	17.9	55.8	44.1	48.2	−0.085	−0.188
Control Region	916	33.1	20.3	34.1	12.6	67.2	32.9	45.7	0.015	−0.236
rRNAs	2627	21.7	23.5	32.7	22.1	54.4	45.6	43.8	0.203	−0.031
tRNAs	1558	27.9	20.8	28.4	23.0	56.3	43.8	50.9	0.009	0.050
Full genome	16,569	28.2	25.7	28.1	18.0	56.3	43.7	46.2	−0.003	−0.177

## Data Availability

The datasets presented in this study were submitted to The National Center for Biotechnology Information (NCBI) database.

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
