# Peer review of "Characterization and Phylogenetic Analysis of the Complete Mitochondrial Genome of Triplophysa microphthalma"

_biology, 2024, doi:10.3390/biology13080608_

Round 1
Reviewer 1 Report
Comments and Suggestions for Authors
Manuscript by Yang et al. reports the complete mitochondrial genome of the small-eyed whiskered loach species for the first time. By using next generation sequencing technique, the authors successfully characterized complete mitogenome of this species, determined gene annotation and carried out phylogenetic analysis to determine its phylogenetic position among species in the family Cobitidae/ Nemacheilidae. This manuscript has potential for publication, but there are several flaws need to be addressed. One of the main issues with this study is that the authors of this paper still use the obsolete species name, Barbatula microphthalma. Referring to Eschmeyer's catalog of fishes (2024), as well as Froese and Pauly (2024)/FishBase, the valid name for Barbatula microphthalma is Triplophysa microphthalma. In my opinion, the most significant part of this study is to provide new insight into the phylogenetic position of this species among other species in the family Nemacheilidae and the genus Triplophysa. By erroneously stating the invalid species names in this manuscript B. microphthalma, the authors in my opinion, need to revise the majority sections of this manuscript. The taxonomic history of Triplophysa microphthalma deserves more depth discussion in the introduction section to build a strong background in relation to Barbatula vs. Triplophysa. In addition, the strategy for creating the dataset also needs to be adjusted to, e.g., include more taxa from the genus Barbatula. The discussion section also needs to be revised and discussed based on the results of the latest analysis.
Introduction section, first paragraph: Triplophysa microphthalma is a species belongs to the family Nemacheilidae, why don’t the authors start from this Nemacheilidae instead of Cobitidae?
Introduction section: Not all readers know about this fish. I recommend the authors to add some information about the biological characteristics of this species
Lines 46-49: Is this just assumption or opinion by the authors? Relevant references need to be cited here
2.4. Phylogenetic analysis: the authors should make adjustment regarding strategy to create dataset (what taxa should be incorporated) for phylogenetic analyses after resolving Barbatula microphthalma vs. Triplophysa microphthalma
2.4. Phylogenetic analysis: what sequence data for phylogenetic reconstruction? Nucleotide sequences? Amino acid sequences? Or both?
2.4 Phylogenetic analysis: what was the best evolutionary model for each dataset?
Lines 130 and 132: where are the citations?
2.4 Phylogenetic analysis: where are the details of setting for RAxML? What taxa were used as outgroups?
2.4 Phylogenetic analysis: why did the authors use rapid bootstrap analysis? Not heuristic search?
Figure 4: I recommend the authors to use rectangular tree layout rather than polar tree layout to be more reader friendly
3.4 Phylogenetic relationships, lines 234-240: Since the valid species name of the small-eyed whiskered loach is Triplophysa microphthalma, it is not surprising that phylogenetic analysis shows the placement of this species is more closely related to other Triplophysa species
3.4 Phylogenetic relationships, lines 241-247: covariance analysis has never been mentioned in Materials and Methods section
4. Discussion, lines 289-297: These sentences are just assumption and opinion by the authors of this paper in my opinion
4. Discussion: this section needs to be carefully revised, especially on the phylogenetic placement of this species, after the authors resolve Barbatula microphthalma vs. Triplophysa microphthalma issue and make considerable revision in the phylogenetic analysis
I am willing to check the revised version of the manuscript when it is ready.
Author Response
Dear Professor,
We are pleased to submit our revised article entitled:”Characterization and Phylogenetic Analysis of the Complete Mitochondrial Genome of Barbatula microphthalma”. We have responded to all suggestions and requirements raised during this round of the peer-review process. We have made the necessary revisions to the manuscript where appropriate. Enclosed below are our point-by-point responses to the comments and suggestions received.
Thank you for the time and effort you have dedicated to evaluating our revised contribution.
Yours sincerely,
Ping Yang (on behalf of all authors)
Answers to the Editor’s comments and questions
(original comments in bold, answers in a plain text below in light green)
EDITOR
* Comments and Suggestions for Authors:
Manuscript by Yang et al. reports the complete mitochondrial genome of the small-eyed whiskered loach species for the first time. By using next generation sequencing technique, the authors successfully characterized complete mitogenome of this species, determined gene annotation and carried out phylogenetic analysis to determine its phylogenetic position among species in the family Cobitidae/ Nemacheilidae. This manuscript has potential for publication, but there are several flaws need to be addressed. One of the main issues with this study is that the authors of this paper still use the obsolete species name, Barbatula microphthalma. Referring to Eschmeyer's catalog of fishes (2024), as well as Froese and Pauly (2024)/FishBase, the valid name for Barbatula microphthalma is Triplophysa microphthalma. In my opinion, the most significant part of this study is to provide new insight into the phylogenetic position of this species among other species in the family Nemacheilidae and the genus Triplophysa. By erroneously stating the invalid species names in this manuscript B. microphthalma, the authors in my opinion, need to revise the majority sections of this manuscript. The taxonomic history of Triplophysa microphthalma deserves more depth discussion in the introduction section to build a strong background in relation to Barbatula vs. Triplophysa. In addition, the strategy for creating the dataset also needs to be adjusted to, e.g., include more taxa from the genus Barbatula. The discussion section also needs to be revised and discussed based on the results of the latest analysis.
Thank you for evaluation of our manuscript. We have revised the MS accordingly. Please find the point by point responses to the comments and suggestions enclosed below. We greatly appreciate the time and effort you and the other reviewers have dedicated to providing constructive suggestions that have helped improve our MS.
* Introduction section, first paragraph: Triplophysa microphthalma is a species belongs to the family Nemacheilidae, why don’t the authors start from this Nemacheilidae instead of Cobitidae?
We have revised the Introduction section in suggested manner (L: 45 to 60, always referring to the file with tracked changes).
* Introduction section: Not all readers know about this fish. I recommend the authors to add some information about the biological characteristics of this species
We have added a profile of the fine-eyed plateau loach in suggested manner (L: 11 to 18).
* Lines 46-49: Is this just assumption or opinion by the authors? Relevant references need to be cited here
Reference added (L: 60)
.
* 2.4. Phylogenetic analysis: the authors should make adjustment regarding strategy to create dataset (what taxa should be incorporated) for phylogenetic analyses after resolving Barbatula microphthalma vs. Triplophysa microphthalma
We have reviewed the information and included a brief description of the taxonomic history of Triplophysa microphthalma in the Introduction section(L:48 to 55)
. However, as the only publicly available complete mitochondrial genome for the genus Barbatula is that of Barbatula barbatula, we regret to inform that we have not been able to make changes to the taxa included in the analysis (L: 48 to 55)
.
* 2.4. Phylogenetic analysis: what sequence data for phylogenetic reconstruction? Nucleotide sequences? Amino acid sequences? Or both?
We used the nucleotide sequences from the complete mitochondrial genomes of 43 species. The 13 PCGs were analyzed using their corresponding nucleotide sequences as well(L: 136 to 138)
.
* 2.4 Phylogenetic analysis: what was the best evolutionary model for each dataset?
The alternative model used the General Time Reversible (GTR) model, the rate heterogeneity model used the GAMMA model, and the BFGS method (Broyden-Fletcher-Goldfarb- Shanno) to optimize the GTR rate parameters (L: 147 to 149)
.
* Lines 130 and 132: where are the citations?
Reference added (L: 145 and 147)
.
* 2.4 Phylogenetic analysis: where are the details of setting for RAxML? What taxa were used as outgroups?
We have revised this section (L: 139 to 142)
.
* 2.4 Phylogenetic analysis: why did the authors use rapid bootstrap analysis? Not heuristic search?
Rapid bootstrap analysis was chosen over traditional methods due to its significantly faster speed, allowing quick estimation of branch support values. Importantly, the results it yields are as reliable and comparable to those of traditional methods.
* Figure 4: I recommend the authors to use rectangular tree layout rather than polar tree layout to be more reader friendly
We have modified(L: 243).
* 3.4 Phylogenetic relationships, lines 234-240: Since the valid species name of the small-eyed whiskered loach is Triplophysa microphthalma, it is not surprising that phylogenetic analysis shows the placement of this species is more closely related to other Triplophysa species
Thank you for your knowledgeable help in correcting this error, and we have revised the relevant statement accordingly (L: 247 to 262)
.
* 3.4 Phylogenetic relationships, lines 241-247: covariance analysis has never been mentioned in Materials and Methods section
We have added a more detailed description of the covariance analysis in the Material Methods section (2.4)(L: 151 to 154)
.
* 4. Discussion, lines 289-297: These sentences are just assumption and opinion by the authors of this paper in my opinion
Thank you for your meticulous review and valuable comments on our paper. The issue you raised regarding the analysis of tRNA sequences has been carefully considered. We acknowledge that certain statements in the original submission may not have adequately reflected that they are based on existing scientific evidence. To address this concern, the following revisions have been made to the text: (1) Additional references were included to experimental data supporting our statements about tRNA sequences and structures; (2) We have clearly identified speculative statements as hypotheses and cited the literature that supports these hypotheses; (3) Some sentences were rephrased to reduce uncertainty and ensure that all conclusions are well-founded on substantial evidence We believe that these revisions will enhance the accuracy and reliability of our paper. We have attached the revised manuscript and respectfully request a further review(L:302 to 318).
* 4. Discussion: this section needs to be carefully revised, especially on the phylogenetic placement of this species, after the authors resolve Barbatula microphthalma vs. Triplophysa microphthalma issue and make considerable revision in the phylogenetic analysis
We greatly appreciate the insightful comments and suggestions you have provided regarding our manuscript,all of which been taken seriously. The following actions were made: (1) We have conducted a thorough review of the literature and consulted with experts in the field to address the taxonomic distinction between B. microphthalma and T. microphthalma. (2) We have re-examined our phylogenetic analysis, incorporating new data and refining our methods to ensure the most accurate placement of the species in question. (3) The revisions include an updated phylogenetic tree with robust statistical support, which we believe will address the concerns raised. (4) We have revised the relevant section of the manuscript to reflect these changes and have provided a clear explanation of the revisions made in the text.
We have attached the revised manuscript, which includes the updated phylogenetic analysis and the associated discussion. We kindly request that you review these changes and provide further feedback if necessary.We are grateful for the opportunity to improve our work based on your expertise, and looking forward to your further guidance.

Reviewer 2 Report
Comments and Suggestions for Authors
The authors have presented a study where they have defined the mitochondrial genome of the small-eyed whiskered loach, Barbatula microphthalma. Interestingly, they have found that B. microphthalma was more closely related to Triplophysa than Barbatula. This study will be of interest to the readers of the journal. I don’t have any significant concerns regarding the data in the paper and their conclusions. But some minor issues need to be addressed before accepting this paper.
General comments:
In line 149 What does “37 typical animal mitochondrial genes” mean? Including the gene names here would greatly enhance the paper.
Line 188 reference missing
Figure 5: The figure is almost illegible; a figure with better resolution should be submitted.
The conclusion B. microphthalma was more closely related to Triplophysa than Barbatula needs more evidence. Why only 47 genomes were selected for the analysis in figure 4, authors should justify that.
Colinear scatter plots for multiple Barbatula genus should be included.
Author Response
Dear Professor,
We are pleased to submit our revised article entitled:”Characterization and Phylogenetic Analysis of the Complete Mitochondrial Genome of Barbatula microphthalma”. We have responded to all suggestions and requirements raised during this round of the peer-review process. Where appropriate, we have changed the manuscript accordingly. Please find the point-by-point responses to the comments and suggestions enclosed below.
Thank you for your time and effort when evaluating our revised contribution.
Yours sincerely,
Ping Yang (on behalf of all authors)
Answers to the Editor’s comments and questions
(original comments in bold, answers in a plain text below in light green)
EDITOR
Comments and Suggestions for Authors:
* The authors have presented a study where they have defined the mitochondrial genome of the small-eyed whiskered loach, Barbatula microphthalma. Interestingly, they have found that B. microphthalma was more closely related to Triplophysa than Barbatula. This study will be of interest to the readers of the journal. I don’t have any significant concerns regarding the data in the paper and their conclusions. But some minor issues need to be addressed before accepting this paper.
Thank you for positive evaluation of our manuscript. We have revised the MS accordingly. Please find the point by point responses to the comments and suggestions enclosed below. We appreciate the time and effort you and the other reviewers spent with constructive suggestions improving our contribution.
* General comments:
In line 149 What does “37 typical animal mitochondrial genes” mean? Including the gene names here would greatly enhance the paper.
The mitochondrial genomes of animals in general contain 37 genes, and the names of these 37 genes are shown on the genome atlas in Figure 1(L:172 always referring to the file with tracked changes).
* Line 188 reference missing
Reference added (L:200 to 201).
* Figure 5: The figure is almost illegible; a figure with better resolution should be submitted.
It has been changed to an image that meets the resolution requirements (L: 263 to 270).
* The conclusion B. microphthalma was more closely related to Triplophysa than Barbatula needs more evidence. Why only 47 genomes were selected for the analysis in figure 4, authors should justify that.
We selected 43 species from the family Cyprinidae, spanning 11 different genera. Although we aimed to included more species from the genus Barbatula, the only available species is Barbatula Barbatula.
* Colinear scatter plots for multiple Barbatula genus should be included.
As with the above, the only species of the genus Barbatula in which a complete mitochondrial genomes available is Barbatula Barbatula. It is believed that with the ongoing research, more complete mitochondrial genomes of the genus Barbatula will be published
Dear Professor,
We are pleased to submit our revised article entitled:”Characterization and Phylogenetic Analysis of the Complete Mitochondrial Genome of Barbatula microphthalma”. We have responded to all suggestions and requirements raised during this round of the peer-review process. Where appropriate, we have changed the manuscript accordingly. Please find the point-by-point responses to the comments and suggestions enclosed below.
Thank you for your time and effort when evaluating our revised contribution.
Yours sincerely,
Ping Yang (on behalf of all authors)
Answers to the Editor’s comments and questions
(original comments in bold, answers in a plain text below in light green)
EDITOR
Comments and Suggestions for Authors:
* The authors have presented a study where they have defined the mitochondrial genome of the small-eyed whiskered loach, Barbatula microphthalma. Interestingly, they have found that B. microphthalma was more closely related to Triplophysa than Barbatula. This study will be of interest to the readers of the journal. I don’t have any significant concerns regarding the data in the paper and their conclusions. But some minor issues need to be addressed before accepting this paper.
Thank you for positive evaluation of our manuscript. We have revised the MS accordingly. Please find the point by point responses to the comments and suggestions enclosed below. We appreciate the time and effort you and the other reviewers spent with constructive suggestions improving our contribution.
* General comments:
In line 149 What does “37 typical animal mitochondrial genes” mean? Including the gene names here would greatly enhance the paper.
The mitochondrial genomes of animals in general contain 37 genes, and the names of these 37 genes are shown on the genome atlas in Figure 1(L:172 always referring to the file with tracked changes).
* Line 188 reference missing
Reference added (L:200 to 201).
* Figure 5: The figure is almost illegible; a figure with better resolution should be submitted.
It has been changed to an image that meets the resolution requirements (L: 263 to 270).
* The conclusion B. microphthalma was more closely related to Triplophysa than Barbatula needs more evidence. Why only 47 genomes were selected for the analysis in figure 4, authors should justify that.
We selected 43 species from the family Cyprinidae, spanning 11 different genera. Although we aimed to included more species from the genus Barbatula, the only available species is Barbatula Barbatula.
* Colinear scatter plots for multiple Barbatula genus should be included.
As with the above, the only species of the genus Barbatula in which a complete mitochondrial genomes available is Barbatula Barbatula. It is believed that with the ongoing research, more complete mitochondrial genomes of the genus Barbatula will be published

Round 2
Reviewer 1 Report
Comments and Suggestions for Authors
The revised manuscript herein by the authors addressed most of my concerns. After resolving these minor points below, I think the manuscript is ready for indexing.
Figure 1: species name on the mitochondrial map still written as Barbatula microphthalma
Line 330: “post-transal polyadenylation”, did the authors mean post-transcriptional modification by polyadenylation?
Line 348-349: I think this sentence is somewhat confusing. What discrepancy did the authors mean? Is it the phylogenetic position of T. microphthalma among other species in the family Nemacheilidae? Or nesting of T. microphthalma within the genus Barbatula vs. Triplophysa? What did the authors mean by specific genes?
Author Response
Dear Professor,
We are pleased to submit our revised article entitled:”Characterization and Phylogenetic Analysis of the Complete Mitochondrial Genome of Triplophysa microphthalma”. We have responded to all suggestions and requirements raised during this round of the peer-review process. We have made the necessary revisions to the manuscript where appropriate. Enclosed below are our point-by-point responses to the comments and suggestions received.
Thank you for the time and effort you have dedicated to evaluating our revised contribution.
Yours sincerely,
Ping Yang (on behalf of all authors)
Answers to the Editor’s comments and questions
(original comments in bold, answers in a plain text below in light green)
EDITOR
* Comments and Suggestions for Authors:
The revised manuscript herein by the authors addressed most of my concerns. After resolving these minor points below, I think the manuscript is ready for indexing.
Thanks again for your suggestions on our manuscript. We have revised the MS accordingly. Please find the point by point responses to the comments and suggestions enclosed below. We greatly appreciate the time and effort you and the other reviewers have dedicated to providing constructive suggestions that have helped improve our MS.
Figure 1: species name on the mitochondrial map still written as Barbatula microphthalma
We have changed the name of the species in Figure 1(L: 173).
Line 330: “post-transal polyadenylation”, did the authors mean post-transcriptional modification by polyadenylation?
Thank you for your careful review and insightful comments on our manuscript. Upon your suggestion, we have reviewed the context and agree that the term was indeed intended to be "post-transcriptional polyadenylation." This was an oversight on our part, and we apologize for any confusion this may have caused. The process we are referring to is the addition of a poly(A) tail to the 3' end of the mRNA molecule after transcription, which is a common post-transcriptional modification that contributes to mRNA stability and translation efficiency.We will promptly correct this error in the revised manuscript and ensure that all related sections are accurate and consistent(L: 293).
Line 348-349: I think this sentence is somewhat confusing. What discrepancy did the authors mean? Is it the phylogenetic position of T. microphthalma among other species in the family Nemacheilidae? Or nesting of T. microphthalma within the genus Barbatula vs. Triplophysa? What did the authors mean by specific genes?
In previous studies, phylogenetic tree analyses were conducted using protein-coding genes as well as the entire mitochondrial genome. The use of the complete mitochondrial genome may have influenced the taxonomic status of the species. We have corrected this oversight in our manuscript, clarifying that the analysis included all genes within the mitochondrial genome (lines 3100 to 312). We have also revised the section where we previously presented this as an error (line 314)
